# Clinical Outcomes according to Timing to Non Invasive Ventilation Initiation in COPD Patients with Acute Respiratory Failure: A Retrospective Cohort Study

**DOI:** 10.3390/jcm12185973

**Published:** 2023-09-14

**Authors:** Lara Pisani, Gabriele Corsi, Marco Carpano, Gilda Giancotti, Maria Laura Vega, Vito Catalanotti, Stefano Nava

**Affiliations:** 1Alma Mater Studiorum, Department of Medical and Surgical Sciences (DIMEC), University of Bologna, 40126 Bologna, Italy; lara.pisani@unibo.it (L.P.); gabriele.corsi2@studio.unibo.it (G.C.); marco.carpano@studio.unibo.it (M.C.); gilda.giancotti@unibo.it (G.G.); maria.vegapittao@studio.unibo.it (M.L.V.); 2Respiratory and Critical Care Unit, IRCCS Azienda Ospedaliero Universitaria di Bologna, 40138 Bologna, Italy; vito.catalanotti@aosp.bo.it

**Keywords:** NIV, mechanical ventilation, acute respiratory failure, nighttime, daytime, non-working days

## Abstract

Nighttime and non-working days are characterized by a shortage of dedicated staff and available resources. Previous studies have highlighted that patients admitted during the weekend had higher mortality than patients admitted on weekdays (“weekend effect”). However, most studies have focused on specific conditions and controversial results were reported. We conducted an observational, monocentric, retrospective cohort study, based on data collected prospectively to evaluate the impact of the timing of NIV initiation on clinical outcomes in COPD patients with acute respiratory failure (ARF). A total of 266 patients requiring NIV with a time gap between diagnosis of ARF and NIV initiation <48 h were included. Interestingly, 39% of patients were not acidotic (pH = 7.38 ± 0.09 vs. 7.26 ± 0.05, *p* = 0.003) at the time of NIV initiation. The rate of NIV failure (need for intubation and/or all-cause in-hospital death) was similar among three different scenarios: “daytime” vs. “nighttime”, “working” vs. “non-working days”, “nighttime or non-working days” vs. “working days at daytime”. Patients starting NIV during nighttime had a longer gap to NIV initiation compared to daytime (219 vs. 115 min respectively, *p* = 0.01), but this did not influence the NIV outcome. These results suggested that in a training center for NIV management, the failure rate did not increase during the “silent” hours.

## 1. Introduction

Non-invasive ventilation (NIV) is the gold standard treatment for acute respiratory failure (ARF) in some defined groups of patients, according to the current evidence [1]. In particular, NIV represents the fundamental treatment for hypercapnic acute respiratory failure (HARF) due to COPD exacerbation (AECOPD). In the first randomized controlled trial, Brochard et al. showed that NIV significantly reduced the need for endotracheal intubation (ETI) when compared to standard medical treatment among AECOPD patients admitted to an intensive care unit (ICU) [2].

The in-hospital mortality rate is also significantly reduced with NIV [3].

Currently, strong evidence supports the use of NIV in patients who develop AHRF (pH ≤ 7.35, PaCO_2_ > 45 mm Hg) and guidelines suggest that NIV should not be used in COPD patients with compensated hypercapnia in the setting of an exacerbation [1].

Furthermore, having an adequately trained and skilled staff is a key determinant affecting the patient’s outcome after receiving NIV [4] as well as the rapid identification of those patients likely to fail an NIV attempt.

Despite the lack of strong evidence, we can state that night shifts and non-working days are characterized by a shortage of dedicated staff among healthcare personnel, possibly leading to a reduction in patient assistance and a poorer quality of care [5,6].

Previous studies have highlighted that patients admitted from the emergency department during the weekend had a higher mortality than patients admitted on weekdays (the so-called “weekend effect”) [7]. However, most studies have focused on specific conditions or diseases, and controversial results were reported [6].

Thus, there is no consensus regarding the effect of weekend admission, its size, and its possible implications, especially in COPD patients with exacerbation complicated by acute hypercapnic respiratory failure. This holds particularly for patients requiring NIV since a link between the available resources and the type of hospital, on the one hand, and the mortality rate, on the other, has been shown [8].

Therefore, we conducted a study to evaluate the impact of the timing of NIV initiation (“daytime” vs. “nighttime”, “working” vs. “non-working days”) on clinical outcomes in COPD patients with AHRF.

## 2. Material and Methods

This is an observational, monocentric, retrospective cohort study, based on data collected prospectively. The study was approved by the local ethics committee (249.2018 Oss). All the patients signed an informed consent form for collecting clinical data recorded in their clinical chart at hospital admission.

All patients admitted to our unit in a time frame of 24 months (February 2018–February 2020) with a main diagnosis of acute COPD exacerbation according to GOLD criteria were screened. All hypoxic and hypercapnic COPD patients acutely requiring NIV with a time gap between diagnosis of acute respiratory failure and NIV initiation <48 h were enrolled. Our internal written protocol, based on the ERS/ATS recommendation for the use of NIV, included respiratory acidosis (pH < 7.35 while breathing room air); hypoxemia (PaO_2_ < 55 mmHg while breathing room air); respiratory rate >20 breaths/min; and severe dyspnea in the absence of an objectively documented cause, such as pneumonia. Inclusion criteria included a time gap between the diagnosis of acute respiratory failure and NIV initiation <48 h. Obviously, according to the attending physician’s judgment, the internal protocol of NIV initiation criteria could be violated. Patients with an age <18 years, clinical instability with a risk or need for immediate intubation and sleep apnoea–hypopnea syndrome, tracheostomy, cancer, congestive chronic heart failure (left ventricular ejection fraction <40%), morbid obesity (body mass index >35), or with previous domiciliary NIV were excluded. All patients included in the study received maximal medical treatment plus oxygen and NIV.

Patients were then stratified into three groups depending on the time of admission: “daytime” from 8.00 a.m. to 8.00 p.m., “nighttime” from 8.00 p.m. to 8.00 a.m.; “working day” from Monday, 8.00 a.m. to Friday, 8.00 p.m.; “non-working day” from Friday, 8.00 p.m. to Monday, 8.00 a.m., plus any public holiday, according to the calendar of public holidays of Italy.

The baseline demographic and clinical data included age, gender, body mass index, prior use of long-term oxygen therapy (LTOT), Charlson Comorbidity Index, and Simplified Acute Physiology Score (SAPS) II score. Other physiologic variables, including respiratory rate, blood pressure, heart rate, arterial blood gases (PaO_2_, PaCO_2_), pH, and PaO_2_/FiO_2_ were recorded immediately after admission to the unit. Finally, the rate and time of endotracheal intubation (ETI), mortality, the interval between the onset of respiratory failure and NIV initiation (hours), NIV duration, and length of hospital stay were also recorded. NIV was delivered in pressure support or pressure-controlled mode with a full-face mask using ICU ventilators in the NIV mode or specifically designed for NIV. The patients’ heart rates, electrocardiograms, SaO_2_, and blood pressures were monitored continuously. Additional oxygen was administered to achieve a SaO_2_ > 92%. Patients received ventilation with a level of pressure support (15.5 ± 2.8 cm H_2_O and an external positive pressure of 4.6 ± 1.6 cm H_2_O) that was adjusted to achieve satisfactory blood gases and a respiratory rate of less than 25 breaths/min. During the first 24 h, NIV was delivered until it was well tolerated (16 to 22 h per day), spaced by intervals of spontaneous breathing with oxygen. In the following hours, the duration of NIV was gradually reduced, if tolerated, until the patient reached a level of complete autonomy with arterial blood gases indicating a pH ≥ 7.35. Weaning from NIV was defined as no need for the reinstitution of any form of mechanical support for 48 consecutive hours. 

### 2.1. Unit Organization

Our unit is composed of a 7-bed respiratory intensive care unit (RICU) which is located within the respiratory department, consisting of an additional 20 beds. The RICU is fully equipped for invasive and non-invasive ventilation and monitoring, and deals mostly with patients needing NIV or who are hard to wean from invasive mechanical ventilation. All attendees are pulmonologists with specific training in the practice of mechanical ventilation and critical care procedures such as sedation, insertion of a central venous catheter, cardio-pulmonary resuscitation, and intubation. Nurses also received specific training. 

All the patients enrolled in the present study were those receiving NIV in the RICU.

### 2.2. Daytime and Working Day

During this time frame, the “typical” staff is composed of 2 attendees, 2 respiratory fellows, 3 nurses, 1 bed manager, and one physiotherapist.

Nighttime and non-working day.

During the nighttime hours and non-working days, the number of staff was reduced, and the staff was composed of 1 attendee, 1 respiratory fellow, 3 nurses, and no physiotherapist. They are also in charge of the respiratory unit outside the RICU.

### 2.3. Study Outcomes

This study aimed at understanding the role of the timing of NIV initiation in COPD patients with HARF among three different scenarios: “daytime” vs. “nighttime”, “working” vs. “non-working days”, and “nighttime or non-working days” vs. “working days at daytime”.

The primary outcome was the rate of NIV failure (need for intubation and/or all-cause in-hospital death) among groups. 

The decision to intubate was taken by the attending clinicians according to pre-defined written criteria used in our unit. The major criteria included respiratory arrest, respiratory pauses with loss of consciousness, psychomotor agitation making nursing care impossible and requiring sedation, heart rate below 50 beats per minute with loss of alertness, and hemodynamic instability with systolic arterial blood pressure below 70 mmHg; the development of conditions requiring intubation either to protect the airways or to manage copious tracheal secretions; and the inability of the patient to tolerate NIV.

Secondary outcomes included the interval between the onset of respiratory failure and NIV initiation, the overall length of hospital stay, and the total duration of NIV treatment.

### 2.4. Statistical Analysis

Data are presented as a mean and standard deviation (SD) or median (interquartile range), as indicated.

The Mann–Whitney U test or Student’s *t*-test was applied to assess statistical differences between groups, as appropriate. The chi-squared test was used for testing relationships between categorical variables.

We considered two-sided *p*-values less than 0.05 to be statistically significant. Statistical analysis was performed with Stata/Se 14.2 software (StataCorp, College Station, TX, USA).

## 3. Results

Out of 931 patients admitted to our unit, 665 were excluded for the reasons illustrated in Figure 1. The remaining 266 patients were included in the analysis.

The baseline characteristics for all included patients are reported in Table 1. The large majority of patients initiated NIV as soon as they were admitted to the RICU.

In all the three sub-analyses considered (“daytime” vs. “nighttime”, “working” vs. “non-working days”, and “nighttime OR non-working days” vs. “working days at daytime”), all the subgroups were homogeneous (Table 2), except for a higher PaCO_2_ baseline value and a higher age in the “daytime” subgroup. 

Concerning the primary outcome, as shown in Table 3, the failure rate was similar in all the groups, ranging from as low as 17% (nighttime OR non-working days) to 26% (working days at daytime).

Since the ERS/ATS Guidelines suggested the use of NIV only in patients with COPD and acidosis, we illustrate in Figure 2 the distribution of patients receiving NIV despite having a pH > 7.35, according to the three groups.

Overall, 104/266 (39%) of them had a compensated hypercapnia (pH = 7.38 ± 0.09 vs. 7.26 ± 0.05 in the acidotic group, *p* = 0.003) at the time of NIV initiation, and no significant differences were observed in the distribution among the three subgroups, as well as for the primary outcome. In particular, the overall rate of NIV success was 24/104 (23%) and 33/162 (20%) for patients with compensated hypercapnia and acidotic patients, respectively.

As shown in Table 4 concerning the secondary outcomes, the only statistically significant finding in the “daytime” vs. “nighttime” subgroups was related to the time to NIV initiation from admission to our unit. In the night hours, the application of NIV was delayed for about 90 min compared to daily hours (219 min vs. 115 min, respectively, *p* = 0.01), but this had no influence on the NIV outcome.

## 4. Discussion

In this single-center observation trial, we have described for the first time the outcomes during weekdays or off-hours of COPD patients admitted with acute hypercapnic respiratory failure and requiring NIV. 

No differences were found in the intubation rate or mortality among three different scenarios: “daytime” vs. “nighttime”, “working” vs. “non-working days”, and “nighttime or non-working days” vs. “working days at daytime”. This study also highlights the fact that in a real-life scenario, that is outside a randomized or observation prospective study, the attending physicians do not strictly follow the recommendations of internal or international guidelines since >30% of the patients enrolled were not acidotic at the time of NIV administration.

The issue of whether off-hours may affect patient outcomes has been widely debated and assessed in several studies in different pathologies and conditions suggesting a worse outcome compared to working days. 

Pauls et al., for example, in a meta-analysis enrolling >50,000 patients, found that weekend admissions had significantly higher overall mortality (relative risk, 1.19) vs. working day admissions, and that no differences were observed among the two groups in the severity of disease, staff numbers, and delays in procedures [9]. Similar data were reported across four different countries (Australia, the United Kingdom, Holland, and the USA) using electronic data [10].

There is, however, concern that these results may be at least partly due to data artifact casting doubt on the use of measures such as hospital standardized mortality rates or retrospective data collected using diagnostic codes and not always representing the severity of the underlying condition [11]. 

Indeed, some other studies demonstrated a different mortality effect according to the underlying pathologies. Zhou et al., for example, demonstrated in another meta-analysis that off-hour admission to the hospital was associated with overall worse outcomes for 28 pathologies, but this association varied greatly across the different diseases [12].

The exact reason(s) for this increased risk of worse outcomes vs. working days are not fully explained but suboptimal standards of care due to a decrease in staff–patient ratios, and reduced diagnostic or therapeutical actions [13,14], are likely to be associated with higher error rates [15]. In particular, patients admitted to the hospital on weekends and requiring an early procedure were less likely to receive the appropriate care in due time [16,17,18]. Other confounders in data interpretation may be also related to a lack of specific descriptions of the type of hospitals (urban vs. rural, teaching vs. non-teaching, and large vs. small), or differences in the severity of illness and/or comorbidities between working and non-working days or day and night shift [19,20], and these potential differences may be a factor in increases in mortality for weekend patients due to a selection bias for weekend versus weekday patients [21]. Among the different pathologies, the off-hours effects on mortality have been significantly demonstrated in several conditions such as myocardial infarction [22,23], stroke [24], upper gastrointestinal hemorrhage [25], hip fracture [26], severe community-acquired pneumonia [27], and pulmonary embolism. Other studies, however, have shown that the weekend or night effect does not apply to all diagnoses [6,28,29].

The issue of COPD, which accounts for approximately 10% of all hospitalizations [30], usually increasing up to 20% during the weekends [31], has also been investigated in a few studies, based on hospital databases and retrospectively collected coding. A Spanish study on almost 300.000 patients with an acute exacerbation of COPD showed significantly higher in-hospital mortality during weekend days (12.9%) than weekday admissions (12.1%) [32], confirming a UK investigation performed in 30 units, demonstrating that higher mortality may be associated with lower doctor numbers [8], and a Finnish study [33]. Another study showed that weekend admission was associated with increased mortality in patients in Japan, which may have been influenced by lower implementation of microbiological testing [27]. 

However, further investigations were unable to find any differences in patient outcomes like a retrospective analysis of administrative data from public hospitals, analyzing the 30-day in-hospital mortality in 30,000 patients in Australia [34], and a US study that observed only a reduced rate of discharge from the hospital on weekends [35].

Notably, however, no study with the exception of Barba’s (around 50% of the total hospital admissions) mentioned or quantified the presence or absence of acute respiratory failure as a cause of acute exacerbation of COPD. While a COPD exacerbation requiring hospital admission is followed by a relatively low mortality rate [36], the association with acute respiratory failure increases the chances of death, especially when invasive or NIV is required [37,38,39]. The use of NIV has been shown to reduce the mortality rate and the need for intubation when compared to standard medical treatment [1], or, on the other hand, as equivalent to invasive ventilation, reducing at the same time the occurrence of complications [40,41]. The timing of application is a critical factor for NIV success [4,42] and therefore working during weekdays or off-hours, may be associated with different outcomes. 

This is, therefore, the first investigation dedicated to the specific condition of acute respiratory failure and the use of NIV in different hospital conditions.

In our study, the application of NIV during the night shift was significantly delayed by about 90 min compared to daily hours (219 min vs. 115 min respectively, *p* 0.01), but this did not influence the NIV outcome. This may be obviously due to the relatively small period difference, while it has been demonstrated that patients admitted to the hospital on weekends and requiring an early procedure were less likely to receive it within 2 days of admission [43]. Our study therefore did not confirm that working off-hours may be associated with worse outcomes, both in sicker patients (those where NIV was applied according to the international guidelines) and in those without acidosis. The reasons for that apparent discrepancy may be due to several factors. First of all, our staff is well trained and skilled, since all forms of respiratory support have been used for many years. Second, all the patients were receiving NIV in a dedicated unit, fully equipped with invasive and noninvasive monitoring systems and dedicated NIV ventilator platforms or ICU ventilators. Third, even during the night shift and non-working days at least three nurses are present for seven beds, which is very similar to the daytime and working days (there is the additional presence of a dedicated bed manager). Last, the personnel were trained in emergency procedures like endotracheal intubation, sedative administration during NIV, and insertion of central venous line, so that in any case any “external” help from other specialists was needed.

Notably, a peculiar finding of our study was that in real life, even in a so-called expert center, around 40% of the patients receiving NIV did not show acidosis, and therefore according to the ERA/ATS document, they should not have been NIV candidates. This attitude has been already demonstrated by Sinuff et al. [44] since only 67.3% of the clinicians fulfilled the guideline NIV eligibility criteria in the post-guideline phase compared to 62.6% in the pre-guideline phase, and thus without showing any significant changes in their real-life behavior. This phenomenon has been described already in critical care medicine, since in 66 French ICUs at the bedside, clinical guidelines were fully applied only in 24% of patients, while the median compliance rate for the relevant guidelines was 75%. [45]. Guidelines are not meant to replace clinical judgment, especially in a stressful situation for a clinician when facing, for example, a patient with hypoxia and respiratory distress, even without acidosis. Indeed, while guideline development and implementation are complex, changing clinician behavior to fully introduce the application of a guideline is even more challenging [46,47]. Our study indeed shows that the use of NIV in non-acidotic patients is not, as suggested by one study [48], badly tolerated or associated with a worse outcome compared to the acidotic group.

Our study has several limitations. First, it was a single-center study with a relatively small group of patients. Indeed, the use of NIV in non-acidotic patients could be criticized but it reflects the actual real-world situation. 

Finally, another limitation is undoubtedly the lack of information about the degree of severity of COPD and the home treatment regimen; nevertheless, the severity of the patient, in the acute setting, is derived from the two scores used in this study (Charlson Index and SAPS II).

However, the study also has some strengths. It is the first one dealing with a procedure to treat an episode of respiratory distress, while in other fields of medicine this has already been studied [16,17]. 

The study is also not based on a database collected by the hospital or coding, but analyzing patient by patient the clinical characteristics, like for example the arterial blood gases, severity, and disability scores, that are lacking in the majority of larger studies, making the results less prone to biases. Last, it is also the first investigation able to split the three periods of the 24 h (i.e., “daytime” vs. “nighttime”, “working” vs. “non-working days”, and “nighttime or non-working days” vs. “working days at daytime”), without showing any statistical differences. 

In conclusion, this study shows for the first time that NIV outcomes are not associated with a different outcome depending on the moment of initiation (off-hours or working days), suggesting that in a monitored unit and with a trained staff the risk of NIV failure does increase in the “silent hours” of the hospital.

## Figures and Tables

**Figure 1 jcm-12-05973-f001:**
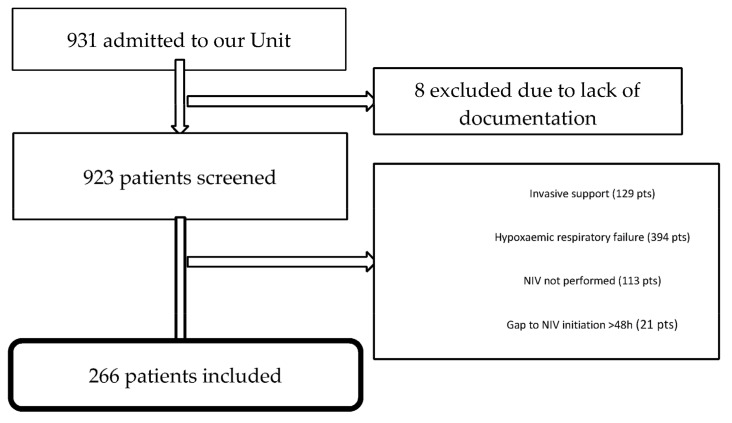
Flowchart of the study.

**Figure 2 jcm-12-05973-f002:**
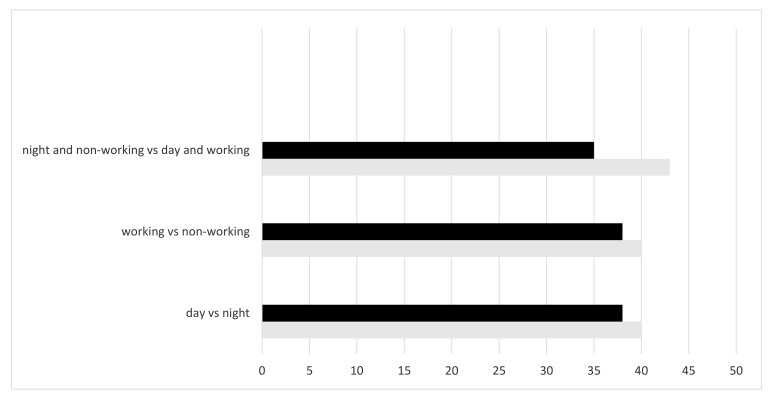
Percentage of patients receiving NIV with a pH ≥ 7.35.

**Table 1 jcm-12-05973-t001:** Subjects’ characteristics on admission to the RICU.

All Included Patients (*n* = 266)	
Age, years	80 (77–81, 95% LCL-UCL)
Sex	Male 114 (42%)
	Female 152 (58%)
pH	7.33 (7.31–7.34, 95% LCL-UCL) 35% absence of respiratory acidosis
Subject’s previous location	231 (86%) Emergency Department 18 (7%) Respiratory ward 9 (4%) Internal medical ward or from other institutions 8 (3%) Intensive Care Unit
LTOT	Yes 130 (48%) No 136 (52%)
NIV at location before RICU admission	No 250 (93%) Yes 16 (7%)

Values are reported as mean +/− standard deviation, if normally distributed; median, minimum, and maximum, if not. Reported percentages for categorical variables. RICU, respiratory intensive care unit; LTOT, long-term oxygen therapy; NIV, non-invasive ventilation.

**Table 2 jcm-12-05973-t002:** Baseline characteristics of all the subgroups “working days” and “non-working days”.

	Nighttime (n = 99)	Daytime (n = 167)	*p*
Age, years	77 (75–81 95% C.I)	81 (78–83 95% C.I)	0.04
sex	M 16.5%/F 20.5%	M 26%/F 37%	0.68
SAPSII score	33 (30–36)	34 (33–36)	0.45
Charlson Index	6 (5–6)	6 (5–6)	0.35
pH	7.33 (7.31–7.35)	7.33 (7.31–7.34)	0.70
PaCO_2_ (mmHg)	65 (63–71)	72 (69–75)	0.02
LTOT	Y 16.5%/N 20.5%	Y 32%/N 31%	0.26
	**Non-Working Days** **(n = 71)**	**Working Days** **(n = 195)**	** *p* **
Age, years	81 (77–83 95% C.I)	79 (77–81 95% C.I)	0.77
sex	M 14%/F 13%	M 29%/F 44%	0.06
SAPS II score	34 (31–38)	34 (32–36)	0.28
Charlson Index	6 (5–6)	6 (5–6)	0.48
pH	7.32 (7.29–7.35)	7.33 (7.31–7.34)	0.97
PaCO_2_ (mmHg)	71 (66–76)	70 (65–73)	0.70
LTOT	Y 15%/N 12%	Y 33%/N 40%	0.20
	**Nighttime OR Non-Working Days** **(n = 137)**	**Working Days at Daytime** **(n = 129)**	** *p* **
Age, years	80 (76–81 95% C.I)	80 (77–84 95% C.I)	0.17
sex	M 24%/F 28%	M 19%/F 29%	0.28
SAPS II score	33 (31–36)	34 (32–36)	0.49
Charlson Index	6 (5–6)	6 (5–6)	0.18
pH	7.33 (7.31–7.34)	7.33 (7.31–7.34)	0.42
PaCO_2_ (mmHg)	69 (64–72)	71 (67–75)	0.17
LTOT	Y 25%/N 26%	Y 24%/N 25%	0.99

Statistical significance for *p*-value < 0.05. Values are reported as mean +/− standard deviation, if normally distributed; median, minimum, and maximum, if not. Reported percentages for categorical variables. LTOT, long-term oxygen therapy; SAPS, simplified acute physiology score.

**Table 3 jcm-12-05973-t003:** NIV success or failure according to the subgroups.

	**Nighttime** **(n = 99)**	**Daytime** **(n = 167)**	*p* 0.27 Pearson value 1.21
Success	81/99 (81%)	127/167 (76%)	
Failure:	18/99 (18%)	40/167(24%)	
Death	8/18 (42%)	15/40(39%)	
Endotracheal intubation	10/18 (58%)	25/40(61%)	
	**nighttime OR non-working days** **(*n* = 137)**	**working days in daytime** **(n = 129)**	*p* 0.08 Pearson value 3.04
Success	113/137 (82%)	95/129 (74%)	
Failure:	24/137 (18%)	34/129 (26%)	
Death	10/24 (40%)	15/34 (42%)	
Endotracheal intubation	14/24 (60%)	19/34 (58%)	
	**non-working days** **(n = 71)**	**working days** **(n = 195)**	*p* 0.19 Pearson value 2.24
Success	59/71 (83%)	149/195 (76%)	
Failure:	12/71 (17%)	46/195 (24%)	
Death	5/12 (42%)	18/46 (39%)	
Endotracheal intubation	7/12 (58%)	28/46 (61%)	

Statistical significance for *p*-value < 0.05. Values are reported as mean +/− standard deviation, if normally distributed; median, minimum, and maximum, if not. Reported percentages for categorical variables.

**Table 4 jcm-12-05973-t004:** Secondary outcomes in the three sub-analyses.

	Nighttime (n = 99)	Daytime (n = 167)	*p*
Gap to NIV initiation (hours)	3.39 ± 8.07	1.55 ± 5.31	0.01
Overall length of NIV treatment (hours)	35 (25–68 95% C.I)	66 (46–86 95% C.I)	0.32
Overall length of stay in hospital (days)	15 (14–17)	16 (15–19)	0.38
	**non-working days**	**working days**	** *p* **
Gap to NIV initiation (hours)	1.86 ± 5.52	2.37 ± 6.8	0.98
Overall length of NIV treatment (hours)	53 (30–87)	48 (37–77)	0.90
Overall length of stay in hospital (days)	16 (14–19)	16 (15–17)	0.76
	**nighttime OR non-working days**	**working days in daytime**	** *p* **
Gap to NIV initiation (hours)	2.65 ± 6.7	2.27 ± 6	0.31
Overall length of NIV treatment (hours)	48 (30–85)	50 (38–77)	0.86
Overall length of stay in hospital (days)	16 (14–17)	16 (15–20)	0.68

Statistical significance for *p*-value < 0.05. Values are reported as mean +/− standard deviation, if normally distributed; median, minimum, and maximum, if not. Reported percentages for categorical variables.

## Data Availability

Not applicable.

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
