# Peer review of "Clinical Outcomes according to Timing to Non Invasive Ventilation Initiation in COPD Patients with Acute Respiratory Failure: A Retrospective Cohort Study"

_jcm, 2023, doi:10.3390/jcm12185973_

Round 1

Reviewer 1 Report

This is an interesting observational study assessing the effect of non-working hours on NIV outcomes in hospitalized patients with COPD.

There are some serious limitations to the design of the study. Importantly, only a small percentage of the study population (7%) initiated NIV after admission to the RICU, while the vast majority (93%) had already initiated prior to their transfer into RICU and inclusion in the study (Table 1). The study, thus, fails to assess staff alertness to NIV timely initiation, but rather evaluates NIV appropriate continuation. This limitation does not support the conclusions drawn.

The authors should consider rephrasing "non-acidotic" patients with mean ph=7.38, to compensated hypercapnia. The authors extensively comment on ERS/ATS guidelines on NIV for acute respiratory failure, but should consider refraining from such commentary or summarizing in one or two sentences, as it is not relevant to the purpose of this observational study.

The authors present hypoxemic respiratory failure as an exclusion criterion in the flowchart/Figure 1. This should become clear in the manuscript as well. The legend of Figure 1 should change to "Patient Selection" or "Flowchart of the study" as it depicts the whole patient selection process, instead of exclusion criteria only. The box containing the exclusion criteria does not allow inspection of the last (or maybe more?) line regarding the number of patients under the 4th bullet.

COPD staging severity and specific comorbidities may affect the outcomes studied here. Consider including these in a table. Since this was an observational study, they could differ between groups potentially affecting the outcomes.

Although the authors mention a primary diagnosis of COPD exacerbation, it is not clear throughout the manuscript, eg in the title (minor comment). Concurrent treatments during hospitalization should be stated to identify potential differences between groups, as these may also affect the outcome.

The results should be more appropriately presented. Regarding the NIV failure outcome, namely death or endotracheal intubation, the authors should report separately the number of patients who died and who were intubated in each group. The timepoints of mortality and intubation should also be clearly stated; were these recorded up to discharge from the hospital, from RICU, or up to a certain timepoint? The authors should clarify whether data were available for all patients included in this study and if there were any missing values.

Line 218: This study was stated as prospective observational, in contrast to the content of this line. 

Line 272: Respiratory failure should not be described as a "cause" for COPD exacerbation. 

Line 272: The term "simple" COPD exacerbation should be erased. Exacerbations are heterogeneous entities according to current literature and research, and this term appears inappropriate. 

Line 328: the phrase "without...difference" should be erased as the study's findings should not be presented in the paragraph regarding limitations and strengths.

There are some typos and minor grammar errors throughout the manuscript (line 15 patient instead of patients, line 329 in conclusions instead of in conclusion, etc.). 

Some sections do not currently make sense and should be rephrased (eg Lines 317-318 "reflecting ... one")

The study's findings should be reported in a past tense, instead of the present tense currently used in some sections (lines 25, 27, etc.).

Author Response

Author’s Reply to the Reviewer 1:

This is an interesting observational study assessing the effect of non-working hours on NIV outcomes in hospitalized patients with COPD.

There are some serious limitations to the design of the study. Importantly, only a small percentage of the study population (7%) initiated NIV after admission to the RICU, while the vast majority (93%) had already initiated prior to their transfer into RICU and inclusion in the study (Table 1). The study, thus, fails to assess staff alertness to NIV timely initiation, but rather evaluates NIV appropriate continuation. This limitation does not support the conclusions drawn.

We are thankful to the Reviewer. We apologize for the misunderstanding, as it was a typing error in Table 1. To demonstrate our “good faith” we want to refer this reviewer to the previous version pag 4 line 169 where we stated “The large majority of patient initiated NIV as soon as they were admitted to the RICU” .We reformulated the text, the large majority of patients initiated NIV at the time of our RICU’s admission (93%).

The authors should consider rephrasing "non-acidotic" patients with mean ph=7.38, to compensated hypercapnia. The authors extensively comment on ERS/ATS guidelines on NIV for acute respiratory failure, but should consider refraining from such commentary or summarizing in one or two sentences, as it is not relevant to the purpose of this observational study.

As for the “non-acidotic” term, we rephrased the term to “patients with compensated hypercapnia”. Concerning the NIV initiation in this type of patients, we respectfully disagree with this reviewer, since we think that this is an important point in our study because it reflects real-life practice and we believe that it is a topic that is intriguing for discussion. It was important to our view to highlight the fact that quite often the International Guidelines, in every field of Medicine are violated in a consistent portion of patients. We do appreciate your feedback.

The authors present hypoxemic respiratory failure as an exclusion criterion in the flowchart/Figure 1. This should become clear in the manuscript as well. The legend of Figure 1 should change to "Patient Selection" or "Flowchart of the study" as it depicts the whole patient selection process, instead of exclusion criteria only. The box containing the exclusion criteria does not allow inspection of the last (or maybe more?) line regarding the number of patients under the 4th bullet.

As for the Figure 1, we modified the Figure’s legend and we modified the box layout so that the 4th bullet is now visible.

COPD staging severity and specific comorbidities may affect the outcomes studied here. Consider including these in a table. Since this was an observational study, they could differ between groups potentially affecting the outcomes.Although the authors mention a primary diagnosis of COPD exacerbation, it is not clear throughout the manuscript, eg in the title (minor comment). Concurrent treatments during hospitalization should be stated to identify potential differences between groups, as these may also affect the outcome.

As for the COPD staging severity, specific comorbidities and concurrent treatments during hospitalization, unfortunately we did not collect these data, but we reported SAPS II score and Charlson Index in table 2  that are representative of the severity of these patients.

The results should be more appropriately presented. Regarding the NIV failure outcome, namely death or endotracheal intubation, the authors should report separately the number of patients who died and who were intubated in each group. The timepoints of mortality and intubation should also be clearly stated; were these recorded up to discharge from the hospital, from RICU, or up to a certain timepoint? The authors should clarify whether data were available for all patients included in this study and if there were any missing values.

Thank you for this suggestion. Regarding the NIV failure outcome, the in-hospital mortality (death) and the intubation rate (endotracheal intubation) are now reported as two separate fields in table 3.

We don’t have the timepoints of mortality and intubation rate because patients have been transferred to other medical units.

We confirm all the data were available and there aren’t missing values.

Line 218: This study was stated as prospective observational, in contrast to the content of this line.

Done  

Line 272: Respiratory failure should not be described as a "cause" for COPD exacerbation. 

Done  

Line 272: The term "simple" COPD exacerbation should be erased. Exacerbations are heterogeneous entities according to current literature and research, and this term appears inappropriate. 

Done  

Line 328: the phrase "without...difference" should be erased as the study's findings should not be presented in the paragraph regarding limitations and strengths.

Done  

Comments on the Quality of English Language

There are some typos and minor grammar errors throughout the manuscript (line 15 patient instead of patients, line 329 in conclusions instead of in conclusion, etc.). 

Some sections do not currently make sense and should be rephrased (eg Lines 317-318 "reflecting ... one")

The study's findings should be reported in a past tense, instead of the present tense currently used in some sections (lines 25, 27, etc.).

As for the comments about the quality of English and syntax, we do appreciate the suggestions and we did some modifications (highlighted in yellow in the text and in the corresponding lines). We have submitted the manuscript to a native speaker for the English review.

We do really appreciate the time you have spent on this review.

Reviewer 2 Report

Thank you for inviting me to review this manuscript on hot topic Clinical outcomes according to timing to noninvasive ventilation initiation in COPD patients with acute respiratory failure: a retrospective cohort study.

The manuscript is easy to read and all data and conclusions presenting huge interest with impact on clinical practice.

I have several proposals:

1.    Can you present more data about COPD patients: severity of obstruction, rate of exacerbations, MRC, GOLD stage?

2.    The idea of silent hours and weekend effect is extremely nice, can you highlight it in the title of the manuscript

Author Response

Thank you for inviting me to review this manuscript on hot topic Clinical outcomes according to timing to noninvasive ventilation initiation in COPD patients with acute respiratory failure: a retrospective cohort study. 

The manuscript is easy to read and all data and conclusions presenting huge interest with impact on clinical practice.

I have several proposals:

  1. Can you present more data about COPD patients: severity of obstruction, rate of exacerbations, MRC, GOLD stage?

Your commentary and suggestions are greatly appreciated. Unfortunately, we do not dispose of data about COPD, such as severity of obstruction, GOLD stage, MRC. This was mostly due because at the time of admission the patients were quite anxious and dyspnoic to undergo even a simple spirometric test.

  1. The idea of silent hours and weekend effect is extremely nice, can you highlight it in the title of the manuscript

Thank you for this suggestion, we modified the title.

We do really appreciate the time you have spent on this review.

Reviewer 3 Report

By my opinion, this is a very interesting study about the clinical outcomes according to timing to non invasive ventilation initiation in COPD patients with acute respiratory failure.

The introduction is well composed and in relation with the study, methods are clear and well described and postulated, results are extensive and informative, and discussion part is adequate.

Conclusions are derived and supported by the results, and references are well and novel.

I suggest only a minor technical correction (to put the title of table 2 before the table, and not within the table legend).

Author Response

By my opinion, this is a very interesting study about the clinical outcomes according to timing to non invasive ventilation initiation in COPD patients with acute respiratory failure.

The introduction is well composed and in relation with the study, methods are clear and well described and postulated, results are extensive and informative, and discussion part is adequate.

Conclusions are derived and supported by the results, and references are well and novel.

I suggest only a minor technical correction (to put the title of table 2 before the table, and not within the table legend).

Your input is very much appreciated. We modified the title of the Table 2.

We are grateful for the time you have spent on this review.